# Transcultural Adaptation of Environmental Health Questionnaire with Attitude, Knowledge, and Skills Scales for Portuguese Nursing Students

**DOI:** 10.3390/nursrep15010013

**Published:** 2025-01-08

**Authors:** Cristina Álvarez-García, Beatriz Edra, Goreti Marques, Catarina Simões, Mª Dolores López-Franco

**Affiliations:** 1Department of Nursing, University of Jaén, 23071 Jaén, Spain; 2Department of Nursing, Santa Maria Health School, 4049-024 Porto, Portugal; beatriz.edra@santamariasaude.pt (B.E.); goreti.marques@santamariasaude.pt (G.M.); catarina.simoes@santamariasaude.pt (C.S.)

**Keywords:** child health, environmental health, nursing students, questionnaire

## Abstract

**Background/Objectives:** Climate change adversely affects some of the fundamental determinants of health, and children are the population group most vulnerable to exposure to environmental risk factors. The main objective of this study was to validate in the Portuguese context three scales to assess attitudes, knowledge, and skills on children’s environmental health. **Methods**: A cross-sectional observational study was developed to translate, adapt, and validate the questionnaire consisting of the following three scales: Attitude Scale (SANS_2), knowledge scale (ChEHK-Q), and skills scale (ChEHS-Q). This was carried out in two phases: the translation and adaptation process and the validation process using classical measure theory and item response theory with undergraduate nursing students. **Results**: We obtained a valid and reliable questionnaire to measure children’s environmental health competence consisting of an attitude scale (α = 0.84), a knowledge scale (Infit = 0.98, Outfit = 0.97, item reliability = 0.98, and people reliability = 0.75), and a skills scale (Infit = 1.00, Outfit = 0.99, item reliability = 0.82, and people reliability = 0.88). The mean score on the attitude scale was 28.15 (5–35) ± 4.61; 14.92 (0–26) ± 4.51 on the knowledge scale; and 42.51 (24–60) ± 6.41 on the skills scale. **Conclusions**: We found that most Portuguese undergraduate nursing students have very good pro-environmental attitudes and good knowledge and skills in dealing with children’s environmental health. The questionnaire obtained in this study will be useful for comparative studies with other countries and for evaluating the effectiveness of educational interventions.

## 1. Introduction

Nowadays, there is no doubt that global climate change is a reality, and different factors affect its occurrence, such as greenhouse gases mainly due to emissions from human activities, melting and loss of glaciers, ocean warming, sea level rise, and other extreme weather events (droughts, floods, and intense tropical cyclones, heat waves, or forest fires) [1].

The World Health Organization (WHO) [2] and The Lancet [3] have stated that climate change will adversely affect some of the fundamental determinants of health: clean air, clean water, sufficient food, and safe housing.

Although the health effects of climate change occur in all people, children are the population group most vulnerable to exposure to environmental risk factors. This is due to their immaturity, psychosocial dependence, and also the lack of positive stimuli and communication in the first three years of life, which will negatively determine the development of physical, mental, and social habits [4,5,6]. Over two-thirds of children’s health is threatened by environmental risks in their homes, where they learn and play [7]. Children under the age of five are particularly vulnerable, as evidenced by the fact that this age group accounts for more than 88% of the diseases caused by climate change [8]. Future generations of children will experience a disproportionate burden of the adverse effects of climate change compared to previous generations. The impact of climate change on the health of this population is wide-ranging, from physical aspects such as increased vulnerability to disease (asthma, allergies, or vector-borne diseases) to other effects that damage mental health (depression or climate anxiety) [9]. The WHO estimates that approximately one-third of the disease burden in developing countries is attributable to modifiable environmental factors, such as indoor and outdoor air pollution, unsafe water, inadequate sanitation, and hygiene [10].

The 2030 Sustainable Development Goals (SDGs) clearly state the need for Education for Sustainable Development (ESD) to sustain planetary health [11]. Similarly, it is necessary to assess environmental health competencies in a global context, as it would allow for the improvement of public health, the development of effective policies, the implementation of global strategies, the promotion of environmental awareness, and the training of competent professionals [12]. It is necessary to include environmental issues in undergraduate nursing education and tools to assess their effectiveness in increasing environmental competencies, attitudes, knowledge, and skills in children’s environmental health, as future nurses will be on the frontline of addressing the health consequences of climate change. This is why it is important to determine environmental competencies in nursing students, and previous research carried out in the Spanish context identified through validated instruments that the attitudes, knowledge, and skills related to children’s environmental health in undergraduate nursing students were not adequate [13].

There are different instruments to determine attitudes, knowledge, and skills on children’s environmental health: the Sustainability Attitudes in Nursing Survey 2 (SANS_2), the Children’s Environmental Health Knowledge Questionnaire (ChEHK-Q), and the Children’s Environmental Health Skills Questionnaire (ChEHS-Q). The skills questionnaire has been used to measure nurses’ attitudes towards the inclusion of climate change and sustainability in the nursing curriculum in other research [14]. Although there are other scales, like the New Ecological Paradigm (NEP) [15], that measure environmental attitudes, they are not specific to nursing. The knowledge and skills questionnaires were developed and used in the Spanish context [16], specifically to measure children’s environmental health knowledge and skills, and to the best of our knowledge, no other tools exist to measure these constructs. These questionnaires were chosen as they are most appropriate for this study population and the topic addressed. In addition, these questionnaires have been adopted in some countries such as Korea [17], China [18], and the United Kingdom [19]; however, there is currently no measurement instrument validated in Portuguese. Therefore, the main objective of this study is to validate in the Portuguese context three scales to assess attitudes, knowledge, and skills on children’s environmental health. The second objective is to determine the knowledge, attitudes, and skills of the Portuguese nursing students in the sample studied.

## 2. Materials and Methods

A cross-sectional observational study was developed to translate, adapt, and validate the questionnaire consisting of the following three scales:

Attitude Scale (SANS_2). This scale was designed to assess nursing students’ attitudes toward sustainability and climate change. It consists of five items, with response options ranging from 1 to 7 on a Likert scale, with a maximum score of 35 points. The score range calculated by the study by Álvarez-García et al. [13] is as follows: Excellent attitude (>90% of perceived attitudes), very good attitude (70–89%), good attitude (50–69%), insufficient attitude (30–49%), and poor attitude (<29%). Reliability analysis of this tool previously revealed a Cronbach’s alpha of 0.82, and the five items loaded on a single factor explained 58% of the variance [14].

Knowledge scale (ChEHK-Q). This scale measures nursing students’ knowledge of children’s environmental health. It consists of 26 items, with true, false, and I don’t know response options. The maximum score on the scale is 26 points. The range of scores is as follows: Excellent knowledge (>90% correct answers), very good knowledge (89–80% correct answers), good knowledge (79–60% correct answers), insufficient knowledge (59–40% correct answers), and poor knowledge (<39% correct answers). This scale shows a good fit and a reliability of 0.98 for items and 0.70 for people based on Rash’s model in the Spanish population [16].

Skills scale (ChEHS-Q). This scale measures the skills of nursing students in children’s environmental health. It consists of 12 items with response options ranging from 1 to 5 on a Likert scale. The maximum scale score is 60 points. The range of scores is as follows: Excellent skills (>90% of perceived skills), very good skills (89–80% of perceived skills), good skills (80–70% of perceived skills), insufficient skills (79–50% of perceived skills), and poor skills (<49% of perceived skills). This scale shows a good fit and a reliability of 0.87 for items and 0.76 for people according to Andrich’s model in the Spanish population [16].

In this study, the guidelines of the International Test Commission [20] were followed for the translation, adaptation, and validation of the questionnaire. The questionnaires are available in Appendix A. This was carried out in two phases:

### 2.1. Phase 1: Translation and Adaptation Process

The questionnaire was translated into Portuguese using a centered or asymmetric method [21], as the questionnaire was intended to be translated without affecting the wording of the original scales. It was decided to do this because the scales were used in the first part of the study conducted in Spain [16] with a similar population of nursing students and have distinctive and desirable characteristics that should remain consistent.

To achieve semantic equivalence (ensure that items have the same meaning in both languages), a translation was carried out by two Portuguese native speakers in the research team. This was followed by a back-translation by two other native Spanish speakers. A matrix was constructed to visualize the original item, the translated items, and the back-translated items. The members of the research team then compared the original and back-translated versions, and the necessary adjustments were made (Appendix A). In order to achieve conceptual equivalence, the translators were members of the research team who are experts in the study construct and faculty members of the institution where the questionnaire will be used, so they are familiar with the target population of the questionnaire.

The translated version of the questionnaire was cognitive questioning with 10 nursing students to assess their opinions on the readability and comprehension of the items [21].

### 2.2. Phase 2: Validation Process

The population used for the psychometric assessment was undergraduate nursing students from a university in northern Portugal. The sample was reached by convenience of those students who wanted to participate in this university, obtaining a sample of 326 students following the methodological recommendations of Polit and Beck (between 5 and 10 subjects) [21] and the assumption that a minimum of 300 subjects are necessary to achieve a robust one-parameter model according to item response theory [22]. The questionnaire was sent by email, the institutional platform, or other digital applications during the academic year 2023–2024.

The psychometric analysis of the scales was intended to be conducted using one-parameter models based on item response theory, which is based on three assumptions [23]. First, the unidimensionality of the scales was tested as suggested by Jiménez-Alfaro and Montero-Rojas [24]. This was conducted through an exploratory factor analysis of each scale and then Horn’s parallel analysis to compare the variance values explained with those obtained in the simulation [25]. Furthermore, at least 50% of the variance needed to be explained, and each extracted factor needed to explain at least 5% of the variance to consider the multidimensionality of the relevant constructs [26]. There also had to be theoretical concordance with the previous known dimensions and the analysis results. Secondly, local independence was also tested using Yen’s Q3 statistic. The recommended values are set at ±0.3 [27]. Thirdly, the Rasch Model analyzed subject (θ) and item (β) parameters on the attitude, knowledge, and skill variables, which will be shown in an item map. The infit and outfit statistics were used to fit the data to the models. Optimal fit values range from 0.7 to 1.3 [28].

Finally, because the attitude scale (SANS_2) did not meet the above assumptions, it was decided to validate according to the classical measurement theory (since it was initially developed under the assumptions of this theory), testing construct validity with an exploratory factor analysis followed by a confirmatory factor analysis and reliability measures such as Cronbach’s α and inter-item correlation with Spearman’s coefficient.

The knowledge scale (ChEHK-Q) was analyzed using the Rasch model. The difficulty of the items was calculated, with 0 being the average level of difficulty. The separation index was also calculated, ideally having values greater than 2 [22].

The ability scale (ChEHS-Q) was analyzed using Andrich’s rating scale model, which is based on the same assumptions and parameters as the Rasch model. In addition to the (mis)fit of the items, this model analyzed the ordering of the thresholds by inspecting the locations of the estimated thresholds along the latent trait, as disordered thresholds may indicate inadequate functioning of the response categories.

For reliability analysis, an internal consistency value of 0.70 or higher was considered acceptable [29].

These analyses were carried out with the jMetrik and JASP 0.18.3.0 software.

### 2.3. Descriptive Analysis

Sociodemographic data of the participants were described. Descriptive values were calculated for the questionnaire scores, the number of hits, and the ignorance index for the knowledge items, and the means for the attitude and skills items. The number of participants with excellent, very good, good, insufficient, or poor attitudes, knowledge, and skills was determined following the guidelines of the instruments. Finally, after checking the normality of the data (*p* > 0.05 in the Kolomogrov–Smirnov test for all the dependent variables), mean difference analyses (ANOVA with Bonferroni post-hoc tests) were carried out to compare the results obtained according to the sociodemographic variables collected. The data were analyzed in JASP 0.18.3.0.

## 3. Results

The results are shown according to the main and secondary objectives marked as headings Section 3.1 and Section 3.2.

### 3.1. Validation in the Portuguese Context Three Scales to Assess Attitudes, Knowledge, and Skills on Children’s Environmental Health

#### 3.1.1. Cognitive Piloting

Cognitive piloting involved 10 students, 8 females (80%) and 2 males (20%); mainly from third year (80%, *n* = 8) or second year (20%, *n* = 2). Most of them had not attended sessions on sustainability and nursing (60%, *n* = 6), although 3 students did (30%) and 1 student did, but more than three months ago (10%).

Regarding the attitude items, the responses were distributed in the established range of 1 to 7 points, with the mean at a high level, reaching a value of 5.87 ± 0.74.

The knowledge items had a mean score of 12.22 ± 3.15 in a range of scores between 9 and 19. The most correct items were 1 and 24, referring to children’s special vulnerability to environmental risks, which were correct for all participants. While the least correct items were 8 and 19, referring to smoking and pesticides, respectively, with a correct score of 22.23%. The rest of the items were distributed in a uniform range of correct answers.

Finally, the skills items obtained a mean score of 3.02 ± 0.34, evenly distributed on the scale of 1 to 5 proposed for these items.

Thus, although there were two knowledge items correct for all participants, no irregular response patterns were detected, as there should always be items in a range from very easy to very difficult to adequately classify respondents, as well as items in the different intervals of the Likert scales in the case of the attitude and skills items.

Post-questionnaire questions showed that the instructions made it clear how to complete the questionnaire, that the meaning of the items was understandable in all cases, and that the format made the questionnaire easy to complete. The completion time ranged from 10 min to 12 min.

#### 3.1.2. Psychometric Study

The total number of students who completed the questionnaire was 326. The mean age was 21.87 ± 6.19 years, ranging between 17 and 56 years. The majority of participants were females 83.44% (*n* = 272) compared to 16.26% (*n* = 53) males and 0.31% transgender (*n* = 1). The highest grade in which the students were enrolled was mostly third grade, with 108 participants (33.13%), followed by second grade with 81 participants (24.85%), 69 first-grade participants (21.17%), and 68 in fourth grade (20.86%). The majority had attended a session on sustainability and nursing, but more than three months ago, 67.49% (*n* = 220), 17.79% (*n* = 58) had not attended any session, and 14.72% (*n* = 48) had attended.

#### 3.1.3. Attitude Scale (SANS_2)

First, an exploratory factor analysis was carried out to check construct validity. Factor analysis was found to be possible, as the Kaiser–Meyer–Olkin score was 0.794, and Bartlett’s test of sphericity was statistically significant (*p* < 0.001). One factor was found to explain 56.10% of the variance, confirming the original unidimensional structure of the scale (Table 1).

Secondly, a confirmatory factor analysis was performed. The factor loadings ranged from 0.29 to 0.93 (Figure 1). The model fit indices were determined as follows: χ^2^ = 33.66, df = 4, *p* < 0.001, RMSEA = 0.137, SRMR = 0.04, GFI = 0.99, CFI = 0.99.

The reliability measured by Cronbach’s α coefficient was 0.84. The inter-item correlation values were lower than 0.20 only for item pairs 2 and 5 (0.18), 4, and 5 (0.18; Appendix A).

#### 3.1.4. Rasch Model of the Knowledge Scale (ChEHK-Q)

First, the unidimensionality of the knowledge scale was demonstrated. For this purpose, it was found that factor analysis was possible for the knowledge scale, as the Kaiser–Meyer–Olkin score was 0.762, and Bartlett’s test of sphericity was statistically significant (*p* < 0.001). In none of the analyses did we obtain good values of variance explained (the factor structure fails to explain 50% of the variance, and two factors fail to explain even 5% of the variance), nor did the items loading on the different components agree with our theoretical assumptions (Table 2); therefore, the scale should be considered unidimensional, as we did not find an internal structure that would justify the contrary.

To test the local independence of the items, Yen’s Q3 correlations were calculated, and the values of the statistics did not exceed the established cut-off value (Appendix A).

Regarding model fit, all 26 items showed central values of Infit and Outfit at 1, confirming the unidimensionality of the construct. The mean Infit was 0.98 (0.85 to 1.23), standardized Infit −0.41 (−3.01 to 4.13), Outfit 0.97 (0.71 to 1.23), and standardized Outfit −0.37 (−2.95 to 2.79). The difficulty values ranged from −2.08 for item 1 (The pediatric population is more susceptible to environmental threats due to their biological immaturity), the easiest item, to 2.78 for item 5 (Nitrogen oxide from fossil fuels in the home and tobacco smoke causes redness and burns on the skin), the most difficult. The item separation index was 7.05, and the person separation index was 1.75. Item separation was robust, but person separation showed some restrictions. The reliability for the item set was 0.98, and for the person 0.75 based on Rash’s model. Again, these values reflect strong item quality but some restrictions in the range of persons in the sample, although both reliability values exceed the established limits. The individual item fit values can be examined in Table 3. To appreciate the performance of the items, they are plotted in Figure 2. In this figure, it can be seen that most of the items are concentrated at intermediate levels of ability, with some items being easier and some more difficult to measure adequately for the higher and lower-ability students in the sample. This could indicate a well-balanced measurement instrument with an appropriate mix of items at different levels of difficulty, allowing one to distinguish between participants of different levels of knowledge.

#### 3.1.5. Rating Scaling Model of the Ability Scale (ChEHS-Q)

The ability scale also proved to be amenable to factor analysis, as the Kaiser–Meyer–Olkin score was 0.885, and Bartlett’s test of sphericity was statistically significant (*p* < 0.001). This was followed by an exploratory factor analysis using principal components as the extraction method. Using Horn’s parallel analysis comparing the explained variance values with those obtained in the simulation, two factors were found (Table 4). The promax oblique rotation was chosen because it calculates the factors from an analytically constructed matrix starting from an orthogonal solution to create a factorial solution as close as possible to the ideal structure. However, no justification was found for the two factors extracted after the rotation, as they only reflected the items with positive and the items with negative sense (Table 5), and the way in which the writing of the items was structured has resulted in spuriously created but theoretically unjustified dimensions. Therefore, the skills scale should also be considered unidimensional.

To test the items’ local independence, Yen correlations were calculated. The values of the Q3 statistic show deviations between items 3 and 6, 2 and 7, 6 and 8, 7 and 11, and 7 and 12. The highest correlation value is shown between items 3 and 6, −0.42. These correlations are detailed in Appendix A.

In the skills scale model, all 12 items showed central values at 1 for Infit and Outfit, confirming the unidimensionality of the construct. The mean Infit was 1.00 (0.76 to 1.47), standardized Infit −0.12 (−2.76 to 4.39), Outfit 0.99 (0.73 to 1.44), and standardized Outfit −0.17 (−3.06 to 3.96). The difficulty values ranged from −0.41 for item 2 (I am NOT able to identify the environmental risks that can cause respiratory diseases in a child), the easiest item, to 0.35 for item 3 (I am able to identify the environmental risks that can cause neoplastic diseases in a child), the most difficult. The item separation index was 2.15, and the person separation index was 2.70. Item and sample separation were robust. The reliability for the set of items was 0.82, and for the person, 0.88. In this case, the reliability values are equally adequate in both cases.

Examining the individual item values (Table 6), item 2 slightly exceeded the maximum values set for Infit and Outfit, without reaching the 1.50 value that would degrade the measure. In order to appreciate the performance of the items, they were represented in Figure 3, which shows that the items are concentrated at intermediate and low levels of ability by fully measuring the students in the sample and classifying them according to their level of ability, leaving out only the outliers. The high ability of the students in the sample can be seen.

The score limits on the Likert scale range from −3.05 to 4.21, increasing in value progressively along the scale, with no setbacks, as shown in Table 7. In addition, the values of Infit and Outfit are in line with the established values, with only the outfit of item 2 showing an under-adjustment.

### 3.2. Determine the Knowledge, Attitudes, and Skills of the Portuguese Nursing Students in the Sample Studied

#### 3.2.1. Descriptive Analysis of the Questionnaire

Table 8, Table 9 and Table 10 below show the responses to the items on attitudes, knowledge, and skills on children’s environmental health.

The final mean score of the attitude scale was 28.15 (5–35) ± 4.61. 94 students (30.62%) had an excellent attitude, 152 students had a very good attitude (49.51%), 55 students had a good attitude (17.92%), 5 students had an insufficient attitude (1.63%), and 1 student had a poor attitude (0.33%).

On the knowledge scale, the mean final score was 14.92 (0–26) ± 4.51. There were 6 students with excellent knowledge (2.18%), 33 students with very good knowledge (12%), 124 students with good knowledge (45.09%), 84 students with insufficient knowledge (30.55%), and 28 students with poor knowledge (10.18%). The ignorance index was higher than 50% only for item 8 (Passive smoking is associated with the development of acute leukemias in children).

The final mean score on the skills scale was 42.51 (24–60) ± 6.41. Twelve students (4.53%) obtained excellent skills scores, 49 students had very good skills (18.49%), 95 students had good skills (35.85%), 101 students had insufficient skills (38.11%), and 8 students had poor skills (3.02%).

#### 3.2.2. Bivariate Analysis of the Questionnaire

In a gendered analysis, male students obtained better environmental health skills (16.41 ± 0.88) than female students (14.93 ± 0.47); *F* = 4.601, *p* = 0.033, η^2^ = 0.04.

Regarding the highest grade in which the student was enrolled, the third-grade student had better attitudes (29.50 ± 0.85, *p* = 0.004) and knowledge (16.10 ± 0.77, *p* = 0.001) compared to the first-grade student with lower attitudes (26.81 ± 1.17) and knowledge (13.14 ± 1.05); *F* = 4.420, *p* = 0.005, η^2^ = 0.04. and *F* = 4.889, *p* = 0.003, η^2^ = 0.05, respectively.

Taking into account the sessions on sustainability and nursing attended by the students, those who had attended more than three months ago had better knowledge (16.51 ± 0.66, *p* = 0.046) but lower skills (41.22 ± 1.02, *p* = 0.018) than the students who had attended recent sessions, with lower knowledge (15.44 ± 0.82) but higher skills (44.32 ± 1.26); *F* = 4.843, *p* = 0.009, η^2^ = 0.03 and *F* = 4.843, *p* = 0.009, η^2^ = 0.018.

## 4. Discussion

The main objective of the research was to translate, adapt, and validate in the Portuguese context three scales to assess knowledge, skills, and attitudes on children’s environmental health in nursing students. The Portuguese versions of ChEHK-Q, ChEHS-Q, and SANS_2 present adequate psychometric properties, providing tools that effectively measure knowledge, skills, and attitudes on children’s environmental health in Portuguese undergraduate nursing students.

The translation, adaptation, and validation of instruments in different languages facilitates safe comparisons if the resulting scales are valid and reliable as well as semantically equivalent [30]. Prior to data collection, cognitive questioning was conducted on a group of participants that was representative of the target population in order to identify problems related to item clarity [31]. In this sense, a rigorous process was followed to obtain instruments that allow for adequate and comparable results over time and across organizations.

In the psychometric analysis of the scales, two types of psychometric theories were used: classical measurement theory and item response theory (Rasch model and Andrich’s model). The knowledge and application of these theories according to the fulfillment of each of the criteria they assume ensures the robustness of the instruments obtained [32].

The use of item response theory techniques converts the raw data to a linear scale that facilitates analysis. The representation of the latent trait of the respondents with the difficulty of the items in the item maps shows the subject-to-subject and item-to-item relationship facilitating the understanding of the results. The model fit parameters for the ChEHK-Q and ChEHS-Q questionnaires in the studied population are considered optimal, except for item 2 of the skills questionnaire, which slightly exceeds the threshold value. Although the analysis of the mapping of the items of the ChEHS-Q questionnaire shows that they are concentrated at intermediate and low levels of ability, the separation index of both items and individuals indicates that the items are adequately distributed in terms of the range of difficulty and the questionnaire is able to detect differences between students with different abilities, just failing to measure outliers. Even so, the separation values found are below those found in studies conducted in China [18]. In the case of the knowledge questionnaire, the separation index of the students is very similar to the values found in research carried out in Spanish [16] and United Kingdom [19] students, with values of 1.53 and 1.75, respectively, below the index considered adequate. The item and person reliability values obtained in the knowledge and skills questionnaire are considered acceptable when reaching the set values (equal to or higher than 0.70) [29], as found in previous research [13,16], with higher reliability in the sample of Chinese students [18].

The values for knowledge (14.92 ± 4.51) and skills (42.51 ± 6.41) of Portuguese students are similar to those found in the study carried out in Spain (knowledge: 15.19 ± 4.08 and skills: 41.10 ± 7.61) [16] and higher than those found in Chinese nursing professionals (knowledge: 13.04 ± 6.27 and skills: 35.28 ± 8.16) [18] and English students (knowledge: 11.59 ± 5.09 and skills: 38.36 ± 7.86) [13]. According to the established cut-off points, and grouping the values into adequate (including the values of excellent, very good, or good) or not adequate (including the values of poor and insufficient), in the Portuguese sample, 40.73% and 41.14% of the students had neither adequate knowledge nor adequate skills, compared to 52.27% and 55.19% of the Spanish students in knowledge and skills respectively, but with better results than those found in English students where only 22.41% and 33.62% had adequate knowledge and skills or the data found in Chinese nursing professionals with values of 19.56% and 19.93% in terms of representation of professionals with good knowledge and skills. The values suggest that nursing professionals possess adequate knowledge and skills. Efforts are therefore required to improve aspects related to environmental health and to comply with professional standards established and defined by some bodies such as the International Council of Nurses [33]. As shown in our study, more general issues, such as those related to vulnerable populations like children or air pollution, are known but more specific pesticides that may cause neoplasms are unknown. Determining the baseline environmental health knowledge and skills of student nurses will enable the designs of educational interventions to improve these variables.

In relation to the SANS_2 questionnaire, both reliability and validity were assessed in the population following classical measurement theory. In order to provide a more valid and reliable factor solution, the exploratory factor analysis took into account two methods for extracting the factors, the choice of those with eigenvalues ˃1 (Kaiser’s criterion), and Horn’s parallel analysis [34]. The results obtained from the factor loadings in terms of construct validity in the confirmatory analysis, although lower in one of them than in other studies conducted by nursing professionals [17] generally indicate an adequate structure for the proposed measurement model. The model fit indices show data that do not reach values within the accepted ranges, such as the case of RMSEA [35]. However, current recommendations indicate the importance of abandoning the practice of applying standard cut-off values indiscriminately to interpret fit indices. It is important to include information on loading factors among other factors related to the model such as construct reliability [36]. On the other hand, the inclusion in the sample of students from different educational levels, including first-year students whose training is limited to basic subjects, may lead to diverse responses that do not adequately reflect the fit in the proposed model. The reliability measured with Cronbach’s alpha indicates acceptable values reaching a value of 0.84, similar to those obtained in other studies of nursing students and professionals [14,17], higher than the value of 0.77 obtained in other research carried out in China [37] or Turkey with 0,75 [38] and below the results found (α = 0.926) in a sample of 159 American nursing students and teachers [39].

Attitude is defined as a disposition to respond favorably or unfavorably to an object, person, institution or event’ [40]. According to the theory of planned behavior, if attitudes affect behavior, nursing students who have a more positive attitude are more likely to engage in sustainable behaviors [41]. The mean score of the attitude scale in Portuguese nursing students was 28.15 ± 4.61, with 80.13% of the surveyed students scoring excellent and good attitude. Several factors appear to influence differences in attitudes, for example, Cruz et al. [42] found significant differences between attitude scale scores in students from Saudi Arabia, Egypt, Iraq, and the Palestinian Territories, and found that country of residence, rural/urban location, and climate change education were variables that could contribute to such differences. Other variables, such as political affiliation, could influence attitude scores [43]. Some research, like ours, indicates the analysis of attitude scores between first-year students and students in more advanced grades, with differences between the two results [39]. This could be because first-year students would not have received training and education in sustainability and environmental issues, which students in higher grades would see included in more advanced and specific subjects. It also seems that the male gender has a higher competence, according to the results of previous studies [19] and the present one, although the samples of male nursing students are very low and this could affect the statistical results found, so this aspect should be further studied, since in most of the studies reviewed there is no analysis with a gender perspective.

The complexity that arises as to how knowledge, attitudes, or skills might influence how people might behave toward issues related to environmental sustainability has led some authors to put forward theories in this regard. The value–belief–norm theory has offered the best explanation for supporting the environmental movement. In this theory, attitudes, environmental knowledge, and values are the central elements in pro-environmental behavior [44]. This behavior will be driven by an individual’s awareness of environmental issues as well as environmental values. Educational interventions that teach about environmental management and sustainability are required so that the users who receive them are not only able to understand but are also able to initiate change [45]. This supports the results of our study, which shows the transformation of learning from environmental sessions into skills and the integration of knowledge in the long term.

In the current literature, there is a growing body of research using educational interventions that significantly increase attitudes toward climate change and sustainability [42,46,47,48,49]; however, sometimes, although this training involves an improvement in attitudes, it is found that students have difficulties in applying sustainability and addressing unsustainable practices in future work environments [46]. Therefore, this education should reinforce the empowerment and leadership of these future nursing professionals as promoters of change.

In spite of the described strengths of our study, it has limitations because, although this study sample is large, the student body is limited to a single training center. In addition, it was not possible to carry out a test–retest analysis because students had further training in the subject after administering the questionnaire. Due to the training the respondents received during their curricula; a large part of the students surveyed received prior training on sustainability. Therefore, a bivariate analysis by subgroups was carried out for a detailed analysis of their influence. In the future, it is proposed to carry out longitudinal studies or by using control groups without affecting the training stipulated in the curriculum of the surveyed students. In addition, the surveys used were self-reported, which may increase the risk of social desirability bias (in the case of attitudes and skills) or the possibility of confirming answers by consulting sources (for knowledge). To reduce this limitation, the answers were collected during the attendance of the theoretical lessons. Finally, although the translation of the questionnaire could be a limitation, a rigorous process was carried out following the guidelines of the International Test Commission for the translation, adaption, and validation of the questionnaires.

## 5. Conclusions

With this study, we have obtained a questionnaire that assesses attitudes, knowledge, and skills in children’s environmental health in a Portuguese context. This questionnaire will be useful to measure environmental health competence in Portuguese nursing students, to evaluate the effectiveness of training, to identify gaps, and to take them into account in curriculum revisions. Furthermore, these questionnaires will allow future comparative studies with other countries where the questionnaire is already culturally adapted and to evaluate the effectiveness of educational interventions in this area. In addition, we found that most Portuguese nursing students studied have very good pro-environmental attitudes and good knowledge and skills in dealing with children’s environmental health.

As future lines of research, we propose validating this questionnaire in health professionals working in different areas where they have to care for children to assess their competence and implement continuous training in deficient areas. This is vital as health professionals must be trained to be resilient against climate change and protect future generations.

## Figures and Tables

**Figure 1 nursrep-15-00013-f001:**
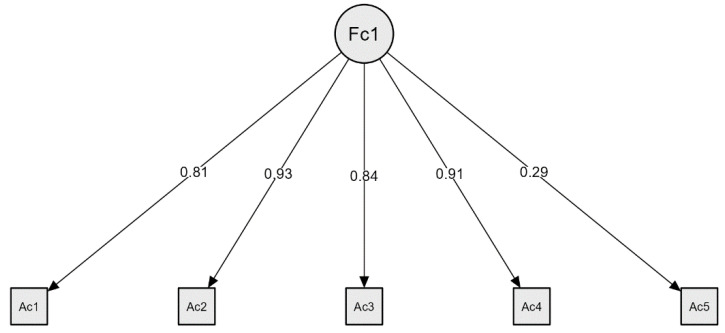
Confirmatory factorial model of the attitude scale.

**Figure 2 nursrep-15-00013-f002:**
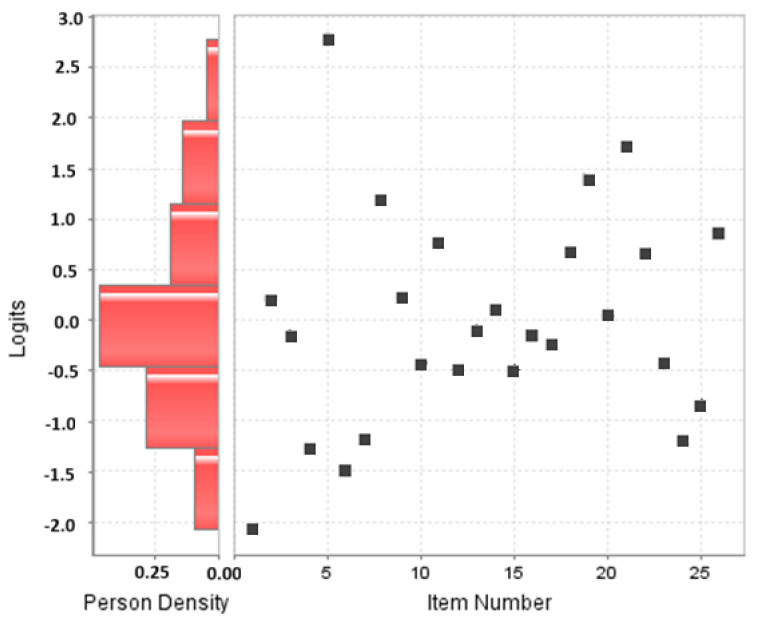
Map of the Rasch model items in the knowledge scale.

**Figure 3 nursrep-15-00013-f003:**
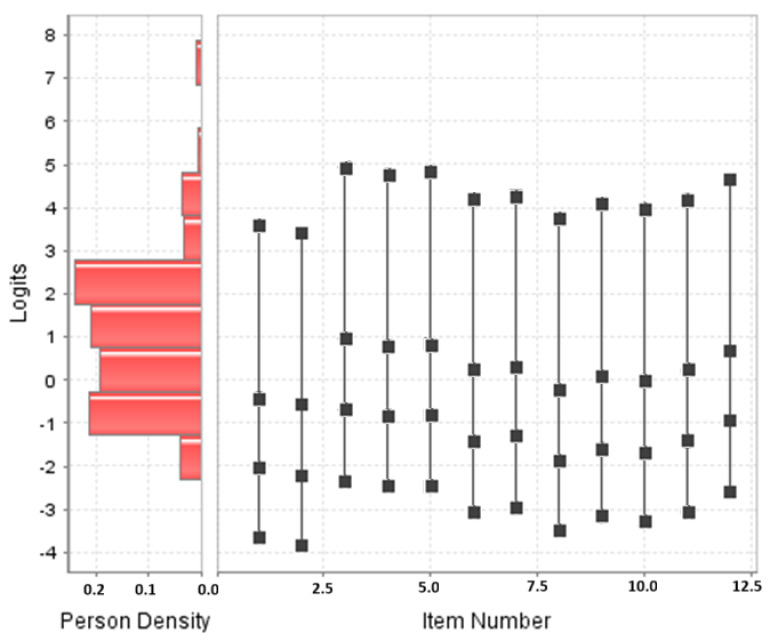
Map of items rating scaling model in the skill scale.

**Table 1 nursrep-15-00013-t001:** Total variance explained and Horn’s parallel analysis of the attitude scale.

Component	Eigenvalues			Parallel Horn Analysis
	Total	% Variance	% Cumulative	Simulation
1 *	3.14	56.10%	56.10%	1.16
2	0.94			1.07
3	0.41			0.99
4	0.35			0.93
5	0.16			0.85

* Factors to be retained after Horn’s parallel analysis.

**Table 2 nursrep-15-00013-t002:** Total variance explained and Horn’s parallel analysis of the knowledge scale.

Component	Eigenvalues			Parallel Horn Analysis
	Total	% Variance		Total
1 *	4.06	12.90%	12.90%	1.56
2 *	2.23	5.80%	18.70%	1.46
3 *	1.53	2.90%	21.60%	1.41
4 *	1.38	2.20%	23.80%	1.35
5	1.27			1.29
6	1.19			1.26
7	1.12			1.21
8	1.06			1.18
9	0.97			1.14
10	0.95			1.11
11	0.93			1.07
12	0.87			1.03
13	0.82			0.99
14	0.78			0.96
15	0.77			0.93
16	0.68			0.90
17	0.66			0.86
18	0.65			0.83
19	0.61			0.80
20	0.58			0.77
21	0.56			0.73
22	0.51			0.70
23	0.50			0.67
24	0.46			0.64
25	0.44			0.61
26	0.42			0.56

* Factors to be retained after Horn’s parallel analysis.

**Table 3 nursrep-15-00013-t003:** Parameters of the Rasch model in the knowledge scale.

Item	Difficulty	Standard Error	Infit: Weighted Mean Square Fit	Infit: Standardized Weighted Mean Square Fit	Outfit: Weighted Mean Square Fit	Outfit: Standardized Weighted Mean Square Fit
1	−2.08	0.21	1.03	0.22	1.07	0.37
2	0.22	0.13	1.05	1.09	1.05	0.79
3	−0.16	0.14	1.23	4.13	1.23	2.79
4	−1.27	0.16	0.87	−1.32	0.71	−2.15
5	2.78	0.20	0.95	−0.33	1.04	0.25
6	−1.49	0.17	1.02	0.26	1.06	0.41
7	−1.19	0.16	1.11	1.17	1.17	1.15
8	1.20	0.14	0.97	−0.47	1.00	0.07
9	0.22	0.13	0.90	−2.28	0.89	−1.64
10	−0.43	0.14	1.02	0.37	1.05	0.57
11	0.77	0.13	1.07	1.53	1.05	0.62
12	−0.50	0.14	0.95	−0.81	0.91	−0.98
13	−0.11	0.13	0.85	−3.01	0.79	−2.95
14	0.11	0.13	0.94	−1.40	0.89	−1.61
15	−0.50	0.14	0.97	−0.39	1.02	0.21
16	−0.16	0.14	0.97	−0.47	0.96	−0.43
17	−0.25	0.14	0.88	−2.21	0.87	−1.61
18	0.67	0.13	0.93	−1.54	0.89	−1.49
19	1.38	0.14	0.96	−0.58	0.90	−0.86
20	0.04	0.13	0.94	−1.34	0.97	−0.40
21	1.71	0.15	0.98	−0.19	0.93	−0.44
22	0.65	0.13	0.92	−1.80	0.87	−1.82
23	−0.43	0.14	0.97	−0.44	0.96	−0.38
24	−1.19	0.16	0.89	−1.23	0.91	−0.57
25	−0.84	0.15	0.94	−0.80	1.02	0.20
26	0.85	0.13	1.05	1.06	1.02	0.23

**Table 4 nursrep-15-00013-t004:** Total variance explained and Horn’s parallel analysis of the skill scale.

Component	Eigenvalues			Parallel Horn Analysis
	Total	% Variance		Total
1 *	5.25	39.70%	39.70%	1.33
2 *	1.61	9.20%	48.90%	1.23
3	0.86			1.17
4	0.72			1.12
5	0.71			1.07
6	0.59			1.01
7	0.48			0.96
8	0.44			0.92
9	0.40			0.87
10	0.36			0.83
11	0.33			0.78
12	0.27			0.72

* Factors to be retained after Horn’s parallel analysis.

**Table 5 nursrep-15-00013-t005:** Factorial loadings of the skills scale.

	Factor 1	Factor 2
8	0.81	
6	0.78	
11	0.74	
12	0.69	
2	0.63	
4	0.42	
7		0.75
5		0.75
9		0.70
10		0.64
3		0.63
1		0.57

Note. The promax oblique rotation method was applied.

**Table 6 nursrep-15-00013-t006:** Parameters of the Rasch model in the skill scale.

Item	Difficulty	Standard error	Infit: Weighted Mean Square Fit	Infit: Standardized Weighted Mean Square Fit	Outfit: Weighted Mean Square Fit	Outfit: Standardized Weighted Mean Square Fit
1	−0.32	0.10	1.04	0.42	0.98	−0.19
2	−0.41	0.11	1.47	4.39	1.44	3.96
3	0.35	0.10	1.22	2.35	1.23	2.28
4	0.28	0.10	0.99	−0.12	1.05	0.55
5	0.29	0.10	0.95	−0.50	0.92	−0.80
6	−0.00	0.10	0.91	−1.00	0.88	−1.31
7	0.04	0.10	0.82	−2.03	0.82	−1.97
8	−0.24	0.10	1.01	0.12	1.01	0.18
9	−0.07	0.10	0.77	−2.68	0.73	−3.06
10	−0.12	0.10	0.89	−1.25	0.87	−1.37
11	−0.01	0.10	0.76	−2.76	0.74	−2.93
12	0.22	0.10	1.16	1.67	1.27	2.66

**Table 7 nursrep-15-00013-t007:** Likert category statistics of the rating scaling model of the ability scale.

Likert Category	Threshold	Standard Deviation	Infit	Outfit
0				
1	−3.05	0.17	1.11	1.30
2	−1.40	0.07	0.71	0.67
3	0.24	0.04	0.80	0.82
4	4.21	0.08	1.30	1.05

**Table 8 nursrep-15-00013-t008:** Responses to sustainability attitude items.

	*n*	Completely Disagree	Disagree	Partially Disagree	Neutral	Partially Agree	Agree	Completely Agree	Mean
1. Climate change is an important issue for nursing.	307	0.32% (1)	0.98% (3)	1.30% (4)	5.86% (18)	14.98% (46)	44.30% (136)	32.24% (99)	5.96
2. Issues about climate change should be included in the nursing curriculum.	307	2.28% (7)	5.86% (18)	3.58% (11)	16.29% (50)	26.06% (80)	30.29% (93)	15.63% (48)	5.11
3. Sustainability is an important issue for nursing.	307	0.33% (1)	0.33% (1)	1.63% (5)	5.54% (17)	18.57% (57)	43.32% (133)	30.29% (93)	5.93
4. Sustainability should be included in the nursing curriculum.	307	1.63% (5)	3.91% (12)	5.21% (16)	13.36% (41)	22.15% (68)	34.53% (106)	19.22% (59)	5.31
5. I apply sustainability principles at home.	307	0.33% (1)	0% (0)	0.98% (3)	6.52% (20)	21.50% (66)	47.56% (146)	23.13% (71)	5.84

**Table 9 nursrep-15-00013-t009:** Children’s environmental health knowledge item scores.

Items	Hits	Ignorance Index
1. The pediatric population is more susceptible to environmental threats due to their biological immaturity.	248/275 (90.18%)	12/275 (4.36%)
2. The increased energy and metabolic consumption of the pediatric population protect children from environmental hazards.	147/275 (53.46%)	77/275 (28.00%)
3. The higher rate of cell growth during the pediatric age increases the risk of health effects caused by environmental factors.	170/275 (61.82%)	63/275 (22.91%)
4. Environmental factors do not influence hormonal secretion during puberty.	224/275 (81.46%)	34/275 (12.36%)
5. Nitrogen oxide from fossil fuels in the home and tobacco smoke causes redness and burns on the skin.	26/275 (9.46%)	78/275 (28.36%)
6. Particles from animals exacerbate asthma crisis.	232/275 (84.36%)	31/275 (11.27%)
7. Increased humidity at home improves respiratory diseases in children.	221/275 (80.36%)	20/275 (7.27%)
8. Passive smoking is associated with the development of acute leukemias in children.	88/275 (32.00%)	140/275 (50.91%)
9. Childhood leukemia incidence rates are higher in the areas most exposed to radon.	147/275 (53.46%)	116/275 (42.18%)
10. Overexposure to solar ultraviolet radiations can damage the skin of adults more severely than that of children.	185/275 (67.27%)	36/275 (13.09%)
11. During childhood, more than half of the expected lifetime solar ultraviolet radiation is absorbed.	113/275 (41.09%)	129/275 (46.91%)
12. Lead accumulates in the body affecting the nervous system.	189/275 (68.73%)	74/275 (26.91%)
13. Chronic dietary exposure to mercury (fish and shellfish) is less toxic to children’s central nervous system than to adults.	167/275 (60.73%)	79/275 (28.73%)
14. Exposure to pesticides increases the risk of developing attention deficit problems in school-aged children.	154/275 (56.00%)	100/275 (36.36%)
15. Children born to smoking mothers during pregnancy are at risk of lower intellectual capacity.	189/275 (68.73%)	39/275 (14.18%)
16. Exposure to organic solvents during fetal development can cause learning disabilities in children.	170/275 (61.82%)	75/275 (27.27%)
17. Water containing nitrates can only cause intoxication during childhood.	175/275 (63.64%)	54/275 (19.64%)
18. Chlorination of water forms sub-products from the disinfection process that have been classified as carcinogenic.	119/275 (43.27%)	120/275 (43.64%)
19. The major source of childhood exposure to pesticides is through ambient air.	78/275 (28.36%)	70/275 (25.46%)
20. The main route of exposure to mercury is through cereal intake.	158/275 (57.46%)	98/275 (35.64%)
21. Exposure to lead through diet occurs mainly through fish intake.	62/275 (22.55%)	97/275 (35.27%)
22. Food colorings and preservatives are associated with central nervous system problems.	120/275 (43.64%)	121/275 (44.00%)
23. Genetically modified foods cause fewer allergic reactions in children.	185/275 (67.27%)	46/275 (16.73%)
24. Schools and nurseries are environmentally safe places.	221/275 (80.36%)	32/275 (11.64%)
25. Children are exposed to higher concentrations of air pollutants at home than outdoors.	206/275 (74.91%)	44/275 (16.00%)
26. Parks and gardens are the areas with the least environmental pollutants where children can play.	108/275 (39.27%)	51/275 (18.54%)

**Table 10 nursrep-15-00013-t010:** Responses to children’s environmental health skill items.

	*n*	Completely Disagree	Disagree	Neutral	Agree	Completely Agree	Mean
1. I am able to assess the main environmental risks to which a child is exposed.	265	1.13% (3)	4.15% (11)	27.55% (73)	59.62% (158)	7.55% (20)	3.68
2. I am NOT able to identify the environmental risks that can cause respiratory diseases in a child.	265	1.51% (4)	8.68% (23)	21.13% (56)	53.96% (143)	14.72% (39)	3.72
3. I am able to identify the environmental risks that can cause neoplastic diseases in a child.	265	1.51% (4)	13.21% (35)	35.47% (94)	45.28% (120)	4.53% (0)	3.38
4. I am NOT able to identify the environmental risks that can cause neurological disorders in a child.	265	1.51% (4)	14.34% (38)	32.83% (87)	43.77% (116)	7.55% (20)	3.42
5. I am able to provide health education to parents about the main contaminants in their child’s food.	265	1.89% (5)	12.83% (34)	33.96% (90)	44.91% (119)	6.42% (17)	3.41
6. I am NOT able to identify the environmental risks in playgrounds.	265	0.76% (2)	11.70% (31)	26.42% (70)	54.34% (144)	6.79% (18)	3.55
7. I am able to provide health education to parents about actions to minimize environmental risks to which a child is exposed when playing outdoors.	265	0.76% (2)	8.68% (23)	32.83% (87)	52.45% (139)	5.28% (14)	3.53
8. I am NOT able to identify the environmental risks in a child’s home.	265	1.13% (3)	7.55% (20)	24.53% (65)	58.87% (156)	7.93% (21)	3.65
9. I am able to provide health promotion to parents about environmental risks at home.	265	0.76% (2)	8.31% (22)	29.82% (79)	54.72% (145)	6.42% (17)	3.58
10. I am able to identify the environmental risks in a child’s school.	265	0.76% (2)	5.66% (15)	30.94% (82)	58.49% (155)	4.15% (11)	3.60
11. I am NOT able to identify the actions needed to combat environmental risks in a child’s school.	265	2.13% (3)	8.68% (23)	32.45% (86)	49.43% (131)	8.30% (22)	3.55
12. I do NOT feel able to do my job as a nurse in a Pediatric Environmental Health Specialty Unit.	265	2.26% (6)	12.83% (34)	34.34% (91)	39.25% (104)	11.32% (30)	3.45

## Data Availability

The original data presented in this study are openly available in FigShare at 10.6084/m9.figshare.27312150.

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
