# Peer review of "Transcultural Adaptation of Environmental Health Questionnaire with Attitude, Knowledge, and Skills Scales for Portuguese Nursing Students"

_nursrep, 2025, doi:10.3390/nursrep15010013_

Round 1

Reviewer 1 Report

Comments and Suggestions for Authors

Dear authors.

My comments and suggestions are as follows:

In the introduction, the authors should indicate the reason for choosing these questionnaires: were they all the ones that were available, or were they the ones that seemed most appropriate? And for what reason?

It should be indicated why only 10 subjects were chosen for the pilot study. Would it not be more convenient to choose a larger sample so that the results obtained would represent the study population? It is usually considered that the pilot study should include between 10 and 20% of the sample size or at least between 30 and 50 subjects.

When indicating a reliability result you should specify the test used (lines 94,101, 240).

To see the inter-item correlation in the SANS_2 questionnaire I consider that it would be more convenient to use Spearman's correlation coefficient since we are dealing with ordinal qualitative variables (Line 147).

It should further justify this statement: “therefore, the scale should be considered unidimensional as we did not find an internal structure that would justify the contrary” (Lines 225-226).

You should specify why the promax oblique rotation was chosen and not an orthogonal rotation (Line 257).        

I do not understand how the questionnaire is said to be unidimensional after the oblique promax rotation if Table 5 shows the existence of two factors.     

I consider that to assess the reliability of the ChEHS-Q questionnaire it would be more convenient to use the McDonald Omega test since it is a Likert questionnaire with fewer than 6 response options (Line 273).

I consider that there are more limitations that should be detailed in addition to external validity. For example, the limitations derived from considering the ChEHK-Q questionnaire unidimensional (Lines 225-226) or the limitations derived from the lack of adjustment of the SANS_2 questionnaire (Lines 387-388).

Kind regards.

Author Response

Comment 1: In the introduction, the authors should indicate the reason for choosing these questionnaires: were they all the ones that were available, or were they the ones that seemed most appropriate? And for what reason?

Response 1: The skills questionnaire has been used to measure nurses' attitudes towards the inclusion of climate change and sustainability in the nursing curriculum in other research and was also assessed psychometrically in several EU countries (England, Spain and Germany). Although there are other scales, like New Ecological Paradigm (NEP), that measure environmental attitudes, they are not specific to nursing. The knowledge and skills questionnaires were developed with reference to the Nursus project which aims to improve the availability of competency-based learning provision on environmental sustainability that can be included in nursing education. These questionnaires have been chosen as they are most appropriate for the study population and the topic to be addressed. This information has been added in the introduction lines 73-81.

Comment 2: It should be indicated why only 10 subjects were chosen for the pilot study. Would it not be more convenient to choose a larger sample so that the results obtained would represent the study population? It is usually considered that the pilot study should include between 10 and 20% of the sample size or at least between 30 and 50 subjects.

Response 2: It was not actually a pilot study, but a cognitive questioning, which was conducted according to Polit and Beck's recommendations, with 10-20 participants. This has been clarified in the text and the reference has been entered on line 135.

Comment 3: When indicating a reliability result you should specify the test used (lines 93,101, 240).

Response 3: In the case of the knowledge scale questionnaire (ChEHK-Q) and ability scale (ChEHS-Q) it was calculated using the Jmetrik program. The reliability is conceptually similar to the cronbach's alpha of classical measurement theory, but uses the expected variance under the Rasch`s and Andrich´s model instead of classical assumptions. This is why it is specified in the case of the knowledge and skills questionnaire that the reliability is obtained according to the Rasch and Andrich model respectively.

Comment 4: To see the inter-item correlation in the SANS_2 questionnaire I consider that it would be more convenient to use Spearman's correlation coefficient since we are dealing with ordinal qualitative variables (Line 147).

Response 4: Thank you very much for your appreciation. It has been corrected (line 165 and Appendix C).

Comment 5: It should further justify this statement: “therefore, the scale should be considered unidimensional as we did not find an internal structure that would justify the contrary” (Lines 225-226).

Response 5: It has been explained in the text that the explained variance is not sufficient to consider the factor structure, and it has no theoretical justification (line 247-249).

Comment 6: You should specify why the promax oblique rotation was chosen and not an orthogonal rotation (Line 257).    

Response 6: The promax oblique rotation was chosen because it calculates the factors from an analytically constructed matrix starting from an orthogonal solution to create a factorial solution as close as possible to the ideal structure. Lines 283-286.

Comment 7: I do not understand how the questionnaire is said to be unidimensional after the oblique promax rotation if Table 5 shows the existence of two factors.

Response 7: Thank you very much for your appreciation. It is true, as you say that two factors appear, however, the two resulting factors grouped on the one hand the affirmative items and on the other hand the negative ones. The way in which the writing of the items has been structured can result in spuriously created but theoretically unjustified dimensions, as in this case. Lines 288-290.

Comment 8: I consider that to assess the reliability of the ChEHS-Q questionnaire it would be more convenient to use the McDonald Omega test since it is a Likert questionnaire with fewer than 6 response options (Line 273).

Response 8: The ability scale (ChEHS-Q) was analyzed using Andrich's rating scale model, which is based on the same assumptions and parameters as the Rasch model. The reliability uses the expected variance under Andrich´s model. This is why no other type of reliability has been used.

Comment 9: I consider that there are more limitations that should be detailed in addition to external validity. For example, the limitations derived from considering the ChEHK-Q questionnaire unidimensional (Lines 225-226) or the limitations derived from the lack of adjustment of the SANS_2 questionnaire (Lines 387-388).

Response 9: Thank you very much for your appreciation. Clarification has been added to the discussion. After revision the only index that does not adjust is the RMSEA. Generally, the SRMR is a robust indicator in different estimation models and is more effective than RMSEA in rejecting models that do not fit closely.  Current recommendations indicate the importance of abandoning the practice of applying standard cut-off values indiscriminately to interpret fit indices. It is important to include information on loading factors among other factors related to the model such as construct reliability (lines 428-432). The ChEHK-Q is considered unidimensional after a detailed factor analysis and the reasons given, which we consider a strength for a Rasch analysis.

Reviewer 2 Report

Comments and Suggestions for Authors

Introduction:
Some sentences are long and complex. You could expand the justification to include the relevance of assessing competencies in environmental health within a global context.

Methodology:
You could better explain the criteria for participant selection and how semantic equivalence was ensured in the translation. Clarity in methods increases the study's reliability.

Results:
Include clear interpretations regarding the distribution of data in relation to skill and knowledge levels, and how these results can be used for educational interventions.

Discussion:
Relate the results obtained to findings from other international studies to reinforce the global relevance of the results.

Conclusion:
The conclusion is clear but could be more specific regarding practical implications.

Other Considerations:
The sample includes 265 students, but there are no details on how the sample size was determined. For psychometric validations, the sample size should be justified based on the proportion of items relative to the number of participants (e.g., a rule of 5-10 participants per item). Although the Rasch model was used, the article does not present detailed fit metrics (e.g., infit and outfit mean square), which are essential to evaluate the quality of the items and their fit to the model. The absence of a temporal stability analysis (test-retest) limits the assessment of reliability over time. This is particularly important for scales used in educational contexts.

The article does not mention whether a normality analysis of the data was conducted before applying parametric tests. Information on the distribution of scores is crucial to validate the choice of statistical methods. Comparative analyses between subgroups (e.g., gender, academic year) were not performed. These analyses could provide additional insights into the functionality of the scales in different contexts. Psychometric results are presented in text or tables but could be complemented with graphs (e.g., Wright maps for the Rasch model) to facilitate interpretation.

Author Response

Comment 1: Introduction: Some sentences are long and complex. You could expand the justification to include the relevance of assessing competencies in environmental health within a global context.

Response 1: The introduction has been rewritten according the comment. Also, it has been explained the relevance of assessing competencies in environmental health within a global context. Lines 58-62.

Comment 2: Methodology: You could better explain the criteria for participant selection and how semantic equivalence was ensured in the translation. Clarity in methods increases the study's reliability.

Response 2: Thank you very much for your comment. It has been clarified what semantic equivalence means in lines 125-126. Also, it was explained the participant selection in lines 141-144.

Comment 3: Results: Include clear interpretations regarding the distribution of data in relation to skill and knowledge levels, and how these results can be used for educational interventions.

Response 3: The interpretation has been included in lines 272-274 and 311-312 for knowledge and skill distribution in the data sample, respectively. To explain how the results can be used for educational interventions, lines 417-419 have been added.

Comment 4: Discussion: Relate the results obtained to findings from other international studies to reinforce the global relevance of the results.

Response 4: A new recent study on the reliability of the SANS questionnaire has been added. Line 438.

Comment 5: Conclusion: The conclusion is clear but could be more specific regarding practical implications.

Response 5: A clarification has been added to lines 498-499.

Comment 6: Other Considerations: The sample includes 265 students, but there are no details on how the sample size was determined. For psychometric validations, the sample size should be justified based on the proportion of items relative to the number of participants (e.g., a rule of 5-10 participants per item). Although the Rasch model was used, the article does not present detailed fit metrics (e.g., infit and outfit mean square), which are essential to evaluate the quality of the items and their fit to the model. The absence of a temporal stability analysis (test-retest) limits the assessment of reliability over time. This is particularly important for scales used in educational contexts.

Response 6: Thank you very much for the clarification. The recommendation of 5-10 subjects is based on the criteria proposed by Polit and Beck. A clarification has been added on line 142-144.The detailed fit metrics of the knowledge are in table 3 and lines 257-260 and for the skills scale are in table 6 and lines 297-300. Temporal stability analysis (test-retest) could not be carried out because the students received training on environmental health after the questionnaires were completed. This limitation has been added to lines 480-482.

Comment 7: The article does not mention whether a normality analysis of the data was conducted before applying parametric tests. Information on the distribution of scores is crucial to validate the choice of statistical methods.

Response 7: This information was clarified in lines 184-185.

Comment 8: Comparative analyses between subgroups (e.g., gender, academic year) were not performed.

Response 8: Bivariate analysis by gender and academic year can be found in section 3.4., highlighting which competency variables are affected by gender and academic year.

Comment 9: These analyses could provide additional insights into the functionality of the scales in different contexts. Psychometric results are presented in text or tables but could be complemented with graphs (e.g., Wright maps for the Rasch model) to facilitate interpretation.

Response 9: The map of items showed in the article is similar to Wring map, but derived from the Jmetrik programme, which is the one used to carry out the analyses. These maps are shown in figures 2 and 3.

Reviewer 3 Report

Comments and Suggestions for Authors

Author Response

Comment 1: Line 26: It should be specified that these are undergraduate nursing students and not graduate students.

Response 1: Thank you very much for the appreciation, it has been added on line 26.

Comment 2: Line 77: It should be specified whether the research protocol has been veerified by an ethics commitee. If not, the reason should be given.

Response 2: This clarification is introduced in the specific section ‘Institutional Review Board Statement’ according to the journal's rules.

Comment 3: Line 126: Please specify whether this sample of 326 students corresponds to all undergraduate nursing students at this university.

Response 3: Students who wished to participate in the Portuguese university were included. A clarification has been added to line 141.

Comment 4: Line 170: The presentation of the results is not very clear, particularly in relation to the first objective of the study. I suggest that you reorganize the results to present then better and make them easier to read, by linking then to the study´s two specific objetives.

Response 4: A clarification has been added at the beginning of the results and the headings have been renumbered according to the clarification. Line 191-192.

Comment 5: Line 197-201: Have you tried to separate the responses according to the grade is in which the students were enrolled to get a more accurate picture?

Response 5: The analysis of responses according to academic degree has been done in bivariate analysis and it is showed for the variables where statistical significance was obtained (lines 326-327).

Comment 6: Line 379: And what can we learn from this? Please specify.

Response 6: A clarification has been added to lines 417-419.

Comment 7: Linea 435-436. The section on study limitations should be extended: 1) I think there is an issue in the presentacion of results if there is no distinction between the students´responses according to their grade.

Response 7: In the results due to the various analyses and tables that have been presented, the distinction between the students´ responses according to their grade was showed only where statistical significance has been found.

Comment 8: 2) The translation of the questionnaire could be a limitacion;

Response 8: It has been included as a limitation and how it has been overcome. Lines 491-493

Comment 9: 3) Also, the fact that a majority of the students have taken a section on sustanibility;

Response 9: This has been included as a limitation and possible ways to avoid such a limitation in future studies. Lines 482-487.

Comment 10: 4) You should also discuss the limitations of the three measurement scales you used in your study.

Response 10: It has been included as a limitation and how it has been overcome. Lines 487-491

Round 2

Reviewer 1 Report

Comments and Suggestions for Authors

Dear Authors.

I consider that the manuscript has been sufficiently improved and I have no additional comments or suggestions.

Kind regards.

Reviewer 3 Report

Comments and Suggestions for Authors

Thank you very much for reviewing the manuscript.